# Efficient Realistic Avatar Generation via Model Compression and Enhanced Rendering

## Abstract

In order to integrate digital avatars into people's lives, efficiently generating complete, realistic, and animatable avatars is a very important requirement. However, with the increasing parameter counts and model sizes, efficiency such as training speed and model sizes are challenged when the models are deployed on devices, while the graphical rule-based micro-renderers, which simplify real-world photorealistic mechanisms such as illumination and reflections, are unable to generate photorealistic images. Based on these issues, we propose a two-stage model compression optimization architecture, where the first stage uses our proposed distillation architecture to compress the model, and the second stage uses our proposed generative adversarial renderer to customize its inverse version to the student network to further improve the realism of digital avatars. Specifically, in the knowledge distillation process, multi-scale feature fusion is achieved by concatenating the output features of RandLA-Net and GCN to combine global and local information to better capture the details and contextual information of the point cloud. We construct assisted supervision, which enables point-level supervision by building the graph topology of the entire point cloud. We also propose to feed the extracted point cloud features as latent codes into our well-designed neural renderer to render more realistic facial images. Experiments show that the method not only improves the network performance but also reduces the parameters and computation of the whole network compared to existing SOTA methods, and our method reduces the number of parameters of the teacher model by about 95% and 90% of the computation in knowledge distillation.

## 1 Introduction

Efficiently creating complete, realistic, and animatable digital avatars is crucial for their integration into daily life. However, achieving all these requirements simultaneously is challenging. Monocular avatar creation is not suitable as it struggles with the complex task of reconstructing articulated facial structures and modeling intricate facial appearances. Traditional methods rely on 3D morphing models (3DMMs) (Li et al. (2017); Paysan et al. (2009)) to represent facial geometry and appearance. While useful for various applications, including avatar generation (Cao et al. (2016); Garrido et al. (2016); Ichim et al. (2015)), 3DMMs have limitations in capturing object-specific static and dynamic details like hair, glasses, and fine facial expressions such as wrinkles due to their linear model constraints. In contrast, neural implicit representations (Mescheder et al. (2019); Mildenhall et al. (2021); Park et al. (2019)) excel at capturing finer details such as hair and eyeglasses but require extensive computational resources for pixel rendering. Recently, deformable point-based representations Zheng et al. (2023) have been used for modeling 3D heads. Point clouds offer greater flexibility and versatility compared to meshes, allowing adaptation to accessories like eyeglasses and representation of complex volumetric structures like hair.

However, when the amount of network parameters is too large, the number of point clouds, if increased, will lead to a large amount of computational cost for subsequent optimization and even lead to environmental problems Patterson et al. (2021).Neural network compression techniques like knowledge distillation (Hinton et al. (2015); Romero et al. (2014); Xu et al. (2017)) and pruning (Frankle & Carbin (2018); Han et al. (2015); LeCun et al. (1989); Li et al. (2016)) offer solutions. Knowledge distillation (KD) Hinton et al. (2015) compresses models by transferring features from larger networks to smaller ones. While accurate teacher models seem ideal, research by Cho and

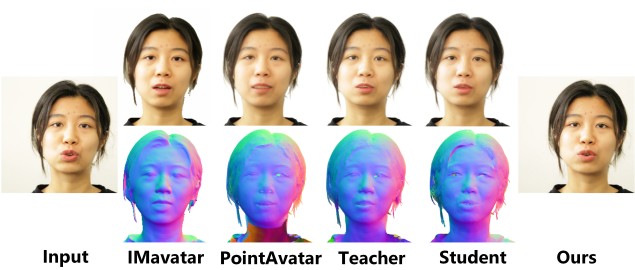

Figure 1: Comparison with state-of-the-art head renderers. The second line of output is the normal map and the first line is the corresponding rendered image. Graphics-based renderer outputs are unrealistic, and our approach faithfully renders a realistic image that agrees with the input.

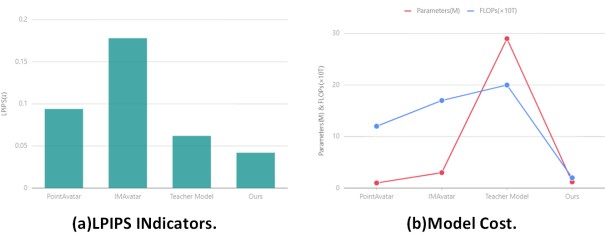

Figure 2: Our method achieves competitive performance while substantially reducing parameters and model size.

Hariharan Cho & Hariharan (2019) suggests simpler teachers can be more effective, especially for smaller student networks (Mirzadeh et al. (2020); Park et al. (2021)). Our enhanced knowledge distillation method features two key improvements: 1) Efficiently building a global point graph topology by concatenating RandLA-Net and GCN output features, enhancing the teacher's model expressiveness through fusion and enabling the student model to benefit from the teacher's knowledge as a second-stage renderer input. 2) Designing auxiliary supervision for disordered point clouds, creating a structured point graph topology for distillation. RandLA-Net handles local point cloud features and context, while GCN addresses global point cloud structure and relationships. This dual approach helps the teacher model learn a richer point cloud feature representation. Point-level supervision enhances the student model's understanding of geometric and semantic features for more accurate representations.

On the other hand, In digital avatar reconstruction, differentiable renderers have been prevalent. However, they have limitations, including unrealistic images due to handcrafted rendering rules and challenges in optimization and training. We propose a self-supervised conditional neural renderer as an alternative to traditional graph-based micro-renderers.

By concatenating RandLA-Net and GCN features, we obtain a rich 3D point cloud representation. This is used as input for a neural renderer based on the generative adversarial network framework, generating realistic 2D images. The renderer is self-supervised, rendering real face images accurately. The optimization process is more stable due to a larger receptive field. As shown in Figure 1, the neural renderer proposed in this paper is able to faithfully render real face images based on the input latent codes. Meanwhile, Inspired by the GAN inversion technique Bau et al. (2019), we design an inversion network to enhance the model's digital avatar performance.

In summary, the main contributions of the proposed method are three-fold:

1. Our enhanced knowledge distillation method optimizes digital avatar generation by improving the teacher model's performance and effectively guiding and optimizing the student network, achieving a novel multi-scale feature fusion and point-level supervision.

2. Our Generative Adversarial Renderer produces highly realistic images and employs Renderer Inverting for further model optimization, resulting in cutting-edge digital avatar realism.

3. Our two-stage compression optimization approach outperforms state-of-the-art methods, offering strong controllability and generating sharp, detailed renderings. Additionally, it reduces the teacher model's parameters by about 95% and computation by 90%, making the student network lightweight and efficient for deployment on devices.

## 2 RELATED WORK

**Head Avatars From 2D.** Creating realistic head avatar from 2D observations is a thriving area in computer vision. Recent progress builds on 3DMMs (Li et al. (2017); Paysan et al. (2009)). Works like (Grassal et al. (2022); Kim et al. (2018)) use neural rendering to capture complex facial appearance and head geometry. This concept extends to generative or disposable avatars in (Bühler et al. (2021); Khakhulin et al. (2022)). Another approach involves neural implicit representations. NerFace Gafni et al. (2021) extends Neural Radiation Fields (NeRFs) Mildenhall et al. (2021) to simulate dynamic head geometries and view-dependent appearances. IMavatar Zheng et al. (2022) uses Neural Implicit Surfaces (Kellnhofer et al. (2021); Mescheder et al. (2019); Park et al. (2019); Yariv et al. (2020)) for generic animation. Implicit avatars are also extended to multi-subject scenes (Bergman et al. (2022); Hong et al. (2022)). pointAvatar Zheng et al. (2023) pioneers point-based deformable head avatars by partitioning RGB colors into albedo and shading. However, these methods have limitations in efficiency, animability, realism, and completeness. Point cloud-based methods excel in drawing and deformation but face GPU memory challenges with large-scale point clouds. Our approach addresses this with two-stage compression optimization after point cloud-based head reconstruction, preventing GPU memory overload.

**Knowledge Distillation.** Knowledge distillation (KD) Hinton et al. (2015) transfers knowledge from a powerful teacher network to a smaller student network. The student network is trained using soft goals and some intermediate features provided by the teacher network (Romero et al. (2014); Yang et al. (2020b); Zagoruyko & Komodakis (2016)).There are variations of KD, such as KD using GAN Xu et al. (2017), Jacobi matching KD (Czarnecki et al. (2017); Srinivas & Fleuret (2018)), activation boundary distillation Heo et al. (2019), comparison distillation Tian et al. (2019), and distillation from graph neural networks (Jing et al. (2021); Yang et al. (2020b)). In recent years, many studies have reported the problem of degradation of student network performance due to the large gap between students and teachers.Cho and Hariharan showed that a network lacking training transfers better knowledge to a smaller network Yang et al. (2020a).Park et al. Park et al. (2021) proposed a student-centered approach to teacher learning in order to efficiently transfer the teacher's knowledge. In this paper, we provide a very simple approach to transfer teachers' knowledge effectively.

## 3 METHOD

Our goal is to compress the model before improving the digital avatar's fidelity. Starting with a monocular RGB video capturing diverse expressions and poses, we generate a color point cloud through PointAvatar. The teacher model processes this point cloud via RandLA-Net and GCN layers, concatenating their output features. We enhance the image and extract image features using Resnet. These two feature sets are fused to generate the final fusion-optimized point cloud using the fusion network. This fusion of features from different domains boosts the model's expressive capacity, benefiting both the teacher and student models. During knowledge distillation, we establish a point-level supervised distillation process through graph topology. Additionally, the concatenated features serve as latent codes for the second-stage generative adversarial renderer, denoted as $G$, which produces more realistic face images. Importantly, $G$ is trained in a self-supervised manner and doesn't rely on labeled data. The renderer $G$ and the renderer inversion network $R$ are fixed after training to further optimize the model using the gap between the inverted output latent codes and the input latent codes to produce more realistic digital avatars. In our experiments, after initially training the renderer $G$ and the renderer inversion network $R$, we alternate between optimization and rendering (both $G$ and $R$) training: every $S$ iterations, we perform 1 step of optimization training and $S$-1 steps of rendering training. In Figure 3, we provide an overview of the methodology.

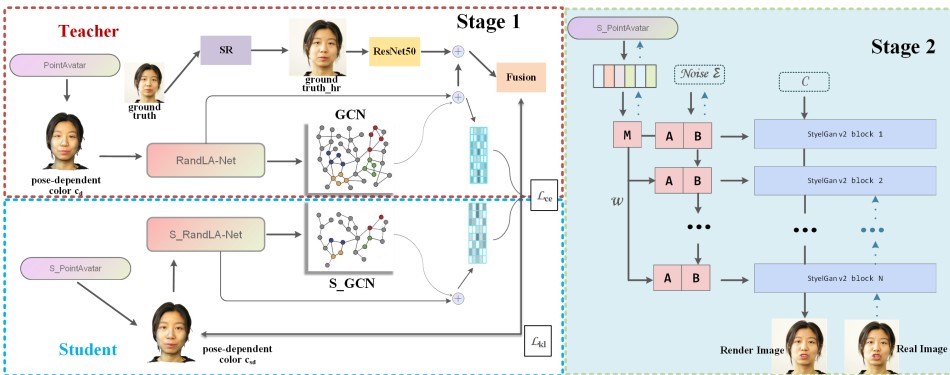

Figure 3: Based on monocular RGB video, we extract the color point cloud by PointAvatar and use RandLA-Net and GCN layers for feature concatenation. Meanwhile, the point-level supervised assisted distillation is realized through graph topology, which integrates the point cloud and image information to enhance the model representation. After renderer G and renderer inversion network R training, we alternate optimization and rendering training to generate more realistic digital avatars.

## 3.1 TEACHER MODEL

In our teacher model, there are two networks: the RandLA-Net Hu et al. (2020) and the Graph Convolutional Network (GCN). For GCN, we choose EdgeConv Wang et al. (2019), where GCN can be represented as:

$$\mathcal{E}_{gcn}(x_i^k) = \mathcal{F}(\{x_i \oplus (x_i - x_i^k) \, | x_i^k \in \mathcal{N}(x_i)\}; \Theta) \tag{1}$$

where $\mathcal{N}(x_i)$ is the local neighborhood of point $x_i$, simply constructed by the $K$ nearest-neighbor algorithm based on the Euclidean distance, and $\oplus$ denotes the feature concatenation. $\mathcal{F}$ is a function with a set of learnable parameters $\Theta$. We then use the Max-pooling operation to aggregate the local features. Thus, a GCN can be represented as:

$$\mathcal{G}(x_i; \Theta) = \max_{x_i^k \in \mathcal{N}(x_i)} \mathcal{E}_{gcn}(x_i^k) \tag{2}$$

In our head avatar creation task, we aim to enhance model generalization and capture finer details. To achieve this, we design the teacher model to fuse 3D point cloud data with 2D image information. This fusion leverages the visual features, such as color and texture, from the image to refine and augment the point cloud data, resulting in higher-quality, more realistic point cloud results. We preprocess the dataset using an image super-resolution strategy, increasing image resolution and detail clarity to aid the network in learning local details. Figure 3 illustrates the process: we first improve image textures with image super-segmentation. Then, we pass the point cloud through RandLA-Net and GCN layers, followed by concatenating their output features. These features are combined with 2D features extracted from ResNet through an MLP. This fusion effectively utilizes information from both the point cloud and image domains, enhancing the model's expressive and generalization capabilities.

## 3.2 KNOWLEDGE DISTILLATION

The improved model after Section 3.1. performs better in the head avatar task, but the increase in parameters and computation leads to a prominent memory footprint problem. To address this difficulty, we adopt a model compression strategy to adapt to the actual deployment requirements and reduce the cost. In order to design a lightweight and efficient student model, we performed compression based on the teacher model. First, the color point clouds are acquired by the compressed S_PointAvatar model, which contains three compressed MLPs.Then, these point clouds are passed through compressed versions of S_RandLA-Net and S_GCN layers to extract and integrate local and global features, respectively. These network layers are streamlined, simplified or merged to reduce the number of parameters and computational complexity. For the knowledge distillation process, we

constructed point-level supervised assisted distillation using graph topology. Specifically, we use the point cloud and point cloud features generated by the teacher model as auxiliary labels and compare them with the output of the student model. By using distribution consistency loss and feature consistency loss, we guide the student model to be consistent with the teacher model in generating the point cloud, while ensuring that the extracted features are as representative of the entire point cloud as possible. In this way, the student model can benefit from the knowledge of the teacher model and gradually learn better representation and optimized point cloud generation. We use distributional consistency loss between the output point clouds: represented as:

$$\mathcal{L}_{kl} = \mathcal{KL}(\mathcal{P}||\mathcal{Q}) \tag{3}$$

Where the point cloud distribution of the teacher model is denoted as $\mathcal{P}$ and the point cloud distribution generated by the student model is denoted as $\mathcal{Q}$. In calculating the loss of distribution consistency, we use the KL scatter as a measure to calculate the difference between the two distributions. $\mathcal{KL}(\mathcal{P}||\mathcal{Q})$ denotes the KL scatter of the teacher model distribution $\mathcal{P}$ with respect to the student model distribution $\mathcal{Q}$, which can be calculated by the following formula:

$$\mathcal{KL}(\mathcal{P}||\mathcal{Q}) = \sum P(x) \log(P(x)/Q(x)) \tag{4}$$

For the point cloud features extracted from the teacher and student models, we use feature consistency loss:

$$\mathcal{L}_{ce} = 1 - (\mathcal{TS})/(\|\mathcal{T}\| \, \|\mathcal{S}\|) \tag{5}$$

Where the output features of the teacher model are denoted as $\mathcal{T}$ and the output features of the student model are denoted as $\mathcal{S}$.

Our distillation method aims to ensure that the student model fully absorbs the knowledge of the teacher model while exploiting its own capabilities, improve the performance of the student model, and reduce the computational load and storage requirements. To this end, we employ a distillation method based on the consistency of the point cloud output distribution and the characteristic cross-entropy loss function to achieve an accurate graph topology for the entire point cloud. Instead of emphasizing the reconstruction accuracy of both models, we focus on ensuring reconstruction consistency between the teacher model and the student model. This method helps to improve the performance of the student model for better applicability, and at the same time ensures that the extracted features are representative of the entire point cloud as a well initialized latent code input to the renderer in the second stage.

### 3.3 GENERATE ADVERSARIAL RENDERER

Our proposed generative adversarial renderer $\mathcal{G}$ is composed of a series of rendering blocks, based on StyleGan v2 Karras et al. (2020), as shown in Figure 3. Each block corresponds to a specific resolution and contains convolutions of varying styles. The modulation and demodulation of the stylized kernel parameter $k$ by the latent code $z$ is defined as:

$$k'_{cij} = w_c k_{cij} / \sqrt{\sum_{c,j} (w_c k_{cij})^2 + \epsilon} \tag{6}$$

$$w = M(z) \tag{7}$$

where $k_{cij}$ is the initial kernel parameter of the $c$-th channel at spatial location $(i, j)$, $k'_{cij}$ is the modulated kernel parameter, and $w_c$ is the modulation parameter of the $c$-th instance channel predicted by the 8-layer MLP $M$ from the latent code $z$ as shown in Equation 7. Here $\epsilon$ is used to avoid dividing numbers by zero. The input feature mapping $f_{cxy}$ is convolved with the modulation kernel to $f'_{lxy}$

$$f'_{lxy} = \sum_{i,j} k^l_{cij} f_{c,x+i,y+j}, \tag{8}$$

where $f'_{lxy}$ indicates the feature map pixel at $(x, y)$ in the $l$-th channel.

In addition to the commonly used GAN loss Goodfellow et al. (2014) that encourages image vividness, perceptual loss Johnson et al. (2016) was used to refine the rendered results, significantly improving detail and realism.

### 3.4 OPTIMIZATION USING RENDERER INVERSION

While fine images with high-frequency details can be obtained using a generative adversarial renderer, it is not enough to generate more realistic 2D images, we also need to use this to optimize our 3D head incarnation. Inspired by Bau et al. (2019), we design a renderer inversion network $R$ that solves this problem with a gradient-based optimization latent code $z$ (as shown by the blue arrow in Figure 3). The renderer inversion network $R$ and the generative adversarial renderer $G$ are trained in a coupled fashion, where the output of the neural renderer (the generated face image $I_{out}$) is fed into $R$ to transform the image back into a latent code $z'$. Ideally, the reconstructed latent code $z'$ should be close to the input latent code $z$. The latent code $z$ is estimated using the statistical mean and variance of the feature mappings, and is given the same as in the style transfer MLP for each layer with the same depth and channel. the resulting feature mapping of the inverse network should have the same spatial size as the feature mapping of the corresponding layer of the neural renderer. The reconstructed latent code $z'$ is estimated based on the concatenation of the statistical mean and standardized variance of each layer, and then based on the MLP. Therefore, the loss function for training the renderer inversion network:

$$
\begin{aligned}
\mathcal{L}_z(R) = &\|\mathrm{MLP}([\mu(R_i(I_{\mathrm{out}})); \sigma(R_i(I_{\mathrm{out}}))]) - z\|_2^2 \\
&+ \sum_i \|G_i(z, \theta) - R_i(I_{\mathrm{out}})\|
\end{aligned}
\tag{9}
$$

where $I_{out} = G(z, noise)$ is the face image generated by the neural renderer, $R_i$ and $G_i$ are the feature maps of the $i$th layer of $R$ and $G$, respectively, and $\mu$, $\sigma$ are the mean and variance of the feature maps in $R$.

### 3.5 TRAINING

**Training strategy:** In our experiments, after initially training the renderer $G$ and the renderer inversion network $R$, we alternate between optimization and rendering (both $G$ and $R$) training (200 iterations for each). When optimizing the 3D avatar avatar, we fix the renderer and the inverse renderer, and for rendering training we fix S_Pointavatar. Figure 4 shows the advantages of this alternating learning.

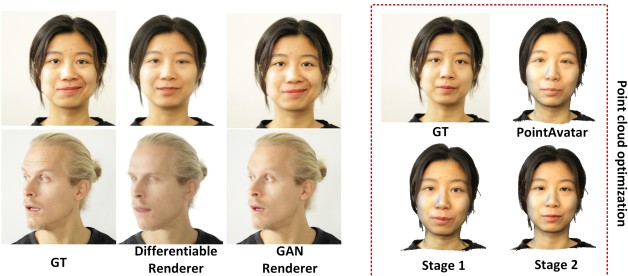

Figure 4: Our generative adversarial renderer produces realistic images while being trained in two stages to effectively enhance the head avatar.

**Losses:** In the first stage, the teacher and student models are reconstructed as 3D head avatars, which are rendered as 2D images. For the 3D head avatars we guide the student model to be consistent with the teacher model in generating the point cloud by using a distribution consistency loss and a feature consistency loss. For 2D images, we follow previous work and use the IMavatar loss:

$$
\mathcal{L}_{\mathrm{vgg}}(\mathbf{C}) = \left\|\mathrm{F}_{\mathrm{vgg}}(\mathbf{C}) - \mathrm{F}_{\mathrm{vgg}}(\mathbf{C}^{\mathrm{T}})\right\|
\tag{10}
$$

$$
\mathcal{L}_{\mathrm{flame}} = \frac{1}{N}\sum_{i=1}^{N}(\lambda_e\|\mathcal{E}_i - \widehat{\mathcal{E}}_i\|_2 + \lambda_p\|\mathcal{P}_i - \widehat{\mathcal{P}}_i\|_2 + \lambda_w\|\mathcal{W}_i - \widehat{\mathcal{W}}_i\|_2)
\tag{11}
$$

$$
\mathcal{L}_{\mathrm{mask}} = \left\|\mathbf{M} - \mathbf{M}^{\mathrm{T}}\right\|
\tag{12}
$$

Here, $\mathbf{C}^T$, $\mathbf{M}^T$, $\widehat{\mathcal{E}}$, $\widehat{\mathcal{P}}$ and $\widehat{\mathcal{W}}$ are obtained from the teacher modeling, and the total loss in our first stage is:

$$\mathcal{L}_{stage1} = \mathcal{L}_{kl} + \lambda_c \mathcal{L}_{ce} + \lambda_v \mathcal{L}_{vgg} + \lambda_f \mathcal{L}_{flame} + \lambda_m \mathcal{L}_{mask} \tag{13}$$

The second stage alternates training optimization and rendering, and the loss function for rendering training is a combination of GAN loss Goodfellow et al. (2014) and perception loss (Dosovitskiy & Brox (2016); Gatys et al. (2016); Johnson et al. (2016)). The loss function for optimizing the 3D head avatar uses feature consistency loss.

## 4 EXPERIMENTS

**Dataset:** To demonstrate the performance of the model, we used the three subject datasets provided by Pointavatar as well as the publicly available dataset from NerFace. One of the NerFace subjects captured videos using a DSLR camera, while the three Pointavatar subjects captured videos using a DSLR camera and a handheld smartphone. We compared our method with the current state-of-the-art (SOTA) method. It is important to note that as handheld smartphones present some limitations in terms of automatic exposure adjustments and image resolution, which pose new challenges for the virtual image approach, our method focuses on addressing these challenging contexts. In all subjects, we used the same facial tracking results and evaluated them in comparison with the individual comparison methods.

**Implementation Details:** Our model was implemented in PyTorch and the first stage was trained by Adam Optimizer optimizer with an initial learning rate of 0.001 and momentum of 0.9. During the second stage of Generative Adversarial Renderer training, we optimized the parameters using Adam solver with a learning rate of 0.01. The first stage trained 60 epochs and the second stage trained 20 epochs. we used 6 attributes (i.e., XYZ coordinates and RGB colors) as inputs for each point, and the entire training time was about 7 hours.

### 4.1 COMPARISON OF METHODOLOGICAL EFFICIENCY WITH EXISTING METHODS

As shown in Table 1, we demonstrate the comparison of our method with the state-of-the-art methods for head avatars. Our method achieves the smallest network parameters and computation. Compared to the teacher model, our parameters and computation are reduced by 95% and 90%, respectively. Also, with a much reduced number of parameters, our method improves accuracy and achieves optimization. This illustrates the effectiveness of our method, especially on the dataset of smartphone shots. As can be seen in Table 1, our method strikes the best balance between reconstruction accuracy, time consistency, and cost compared to Pointavatar. In addition, our method's GPU memory usage is about one-sixth of that of the teacher's model, still the smallest among existing methods.

|  | Parameters(M) | MACs(T) | FLOPs(T) | Runtime Memory(MB) |
|---|---|---|---|---|
| Teacher Model | 29 | 1.02 | 2.05 | 26429 |
| IMavatar | 3 | 0.86 | 1.73 | 6473 |
| PointAvatar | **1** | 0.6 | 1.2 | 4835 |
| Ours | 1.2 | **0.12** | **0.24** | **4017** |

Table 1: Comparison of parametric quantities, computation and Runtime Memory of models.

### 4.2 COMPARISON TO THE STATE OF THE ART

**Quantitative Results.** Table 2 quantitatively juxtaposes our approach with a state-of-the-art baseline (SOTA), utilizing standard metrics such as L1, LPIPS, Zhang et al. (2018), SSIM and PSNR. It is worth noting that in the case of NHA Grassal et al. (2022), its scope is limited to the simulation of the head region; therefore, our comparison with NHA applies only to the head region and excludes the garment domain. Impressively, our method obtained the most favorable metrics across all methods, including lab-captured DSLR camera sequences and handheld smartphones.

**Qualitative Results.** An example of an image rendered by our method can be seen in Figure 4, which demonstrates that our proposed generative adversarial renderer can generate face images of

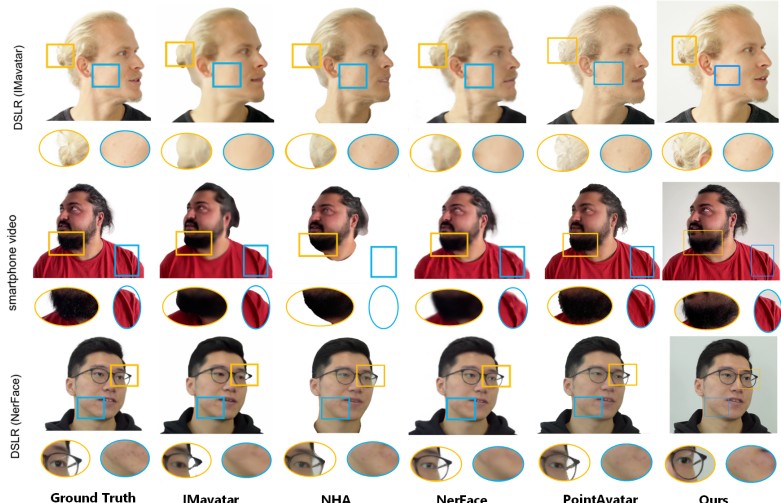

Figure 5: **Qualitative comparison.** Our method produces photo-realistic and detailed appearance compared to SOTA methods, especially apparent in skin details and hair textures.

|  | L1↓ | LPIPS↓ | SSIM↑ | PSNR↑ |
|---|---|---|---|---|
| IMavatar | 0.033; 0.050 | 0.178; 0.261 | 0.874; 0.770 | 22.4; 18.7 |
| NerFace | 0.030; 0.045 | 0.126; 0.187 | 0.877; 0.782 | 22.7; 19.6 |
| PointAvatar | 0.021; 0.036 | 0.094; 0.145 | 0.899; 0.802 | 26.6; 22.3 |
| Ours | **0.015; 0.019** | **0.042; 0.057** | **0.962; 0.931** | **32.7; 28.7** |
| NHA | 0.022; 0.029 | 0.086; 0.123 | 0.890; 0.837 | 25.7; 21.6 |
| PointAvatar (no cloth) | 0.017; 0.021 | 0.077; 0.100 | 0.912; 0.863 | 28.6; 25.8 |
| Ours (no cloth) | **0.011; 0.012** | **0.031; 0.048** | **0.978; 0.969** | **36.8; 35.4** |

Table 2: **Quantitative comparison.** The first and second numbers in each cell represent the scores of videos captured by digital SLRs and videos captured by smartphones, respectively. Our head avatar method highlights superior advantages in quantitative analysis that integrates several key performance metrics.

much higher visual quality than traditional graph-based renderers, with hair, glasses, and other attributes being generated well. Our rendered image is very close to the input image as we significantly reduce the gap between the rendered image and the real image.

Figure 5 shows the qualitative results of our method on smartphone and DSLR data, where our method excels on several fronts, with a particular focus on capturing complex hair details and improving overall image realism. Focusing more on maintaining the integrity of the reconstructed avatar, our method demonstrates the ability to preserve detail, which is particularly evident in the ability to accurately reconstruct hair details, a challenging achievement that is often overlooked by existing methods.

## 4.3 ABLATION STUDY

In this section, we present the results of an ablation study that investigated different components of our proposed two-stage head avatar, and the experimental results are shown in Table 3. We refer to the pre-trained model provided by PointAvatar as "pre-trained". Because the second stage needs to use the extracted point cloud features as latent codes, it is not possible to directly replace the baseline micro-renderer with the generative adversarial renderer, and it is necessary to train a teacher model, but the teacher model cannot be trained alternately in the second stage without compression with our equipment, so the ablation experiment is not considered without the first stage. From the table, we can see that each of the proposed modules and losses are valid.

| Method | L1↓ | | LPIPS↓ | | SSIM↑ | | PSNR↑ | |
|---|---|---|---|---|---|---|---|---|
| Pretrained | 0.024; | 0.029 | 0.098; | 0.100 | 0.874; | 0.821 | 25.1; | 22.8 |
| w/o stage 2 | 0.019; | 0.026 | 0.069; | 0.085 | 0.919; | 0.831 | 26.7; | 23.7 |
| w/o $\mathcal{L}_{ce}$ | 0.016; | 0.021 | 0.047; | 0.065 | 0.932; | 0.901 | 30.6; | 26.3 |
| w/o $\mathcal{L}_{kl}$ | 0.017; | 0.023 | 0.056; | 0.071 | 0.912; | 0.878 | 28.5; | 25.1 |
| w/o initial | 0.094; | 0.098 | 0.384; | 0.396 | 0.484; | 0.425 | 15.6; | 13.1 |
| Ours full | **0.015**; | **0.019** | **0.042**; | **0.057** | **0.962**; | **0.931** | **32.7**; | **28.7** |

Table 3: Ablation study of DSLR and smartphone video.

**Effectiveness of point cloud processing network.** We validate the effectiveness of the point cloud processing network by conducting experiments. Figure 6 (a) presents the results, demonstrating that concatenating the output features of RandLA-Net and GCN in the first stage effectively establishes a global point graph topology and enhances feature representation. However, when using PointNet++ Qi et al. (2017) as the point cloud processing network, the results are unsatisfactory, as it is not suitable for our dynamic point cloud model. PointNet, another option, doesn't perform as well as RandLA-Net and leads to a higher parameter count. Furthermore, Figure 6 (b) reveals that removing GCN diminishes the effectiveness of the model, with only a marginal reduction in parameters.

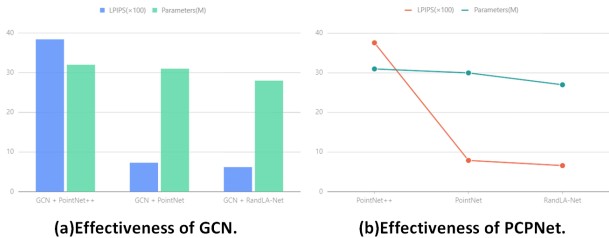

(a)Effectiveness of GCN.   (b)Effectiveness of PCPNet.

Figure 6: Effectiveness ablation study of GCN and point cloud processing networks(PCPNet).

**Effectiveness of distillation losses.** In the process of knowledge distillation, we use graph topology to construct a point-level supervised assisted distillation. We experimentally demonstrate the role of distribution consistency loss and feature consistency loss, as shown in Table 3. Without these two losses, the student model does not align well with the instructor's model in generating the point cloud, which will affect the student model's ability to learn to express itself better.

**Effectiveness of two-stage optimization.** As shown in row 3 and row 4 of Table 3, the student model at the time of distillation that directly uses the generative adversarial renderer instead of the microscopic renderer, which does not have a proper initialization of the latent code, shows a serious increase in reconstruction errors and may not converge for specific facets.

## 5  CONCLUSION

In this paper, we propose a two-stage model compression optimization architecture that achieves significant results in digital avatar generation through a combined approach of knowledge distillation and generative adversarial renderer. Compared to the current state-of-the-art methods, this approach not only improves the network performance, but also drastically reduces the parameters and computation of the whole network. In addition, the effectiveness of the designed distillation loss and point cloud processing network is also validated by the ablation study. The results further indicate that additional supervised and graph topology learning is important for improving the header incarnation of large-scale point clouds.

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

## A    APPENDIX

You may include other additional sections here.

