# OpenReview forum: "Efficient Realistic Avatar Generation via Model Compression and Enhanced Rendering"
_ICLR.cc/2024/Conference — ICLR 2024 Conference Withdrawn Submission_

### Official Review · Reviewer_vN9C · 2023-10-19

**Soundness:** 3 good
**Presentation:** 2 fair
**Contribution:** 2 fair
**Rating:** 3
**Confidence:** 4

**Summary:**

The authors propose two strategies to improve the digital head modeling. The main idea is to use the knowledge distillation with an advanced teacher network, together with improved rendering components. Some results are presented to verify their validness.

**Strengths:**

+ Modeling human heads is critical to a number of down-streamed applications.
+ The authors show some experiment results with quantitative advantages although the qualitative outputs look questionable.

**Weaknesses:**

+ The main idea: enhancing the expressive capability by using multiple domains' information and powerful GANs (e.g. styleGAN v2), and then performing model compression, looks to be trivial. Plus applying knowledge distillation method for compression is also a common idea, I do not see many specific technical insights for the problem the authors want to improve. I am especially looking for some more adaptive designs for the area of human head modeling rather than some combinations of general technical achievements.
+ Considering the ad-hoc nature of the general framework, more ablation studies are needed for the effectiveness of every presented component like RandLA-Net and GCN.
+ The proposed model compression strategy can not only be applied to PointAvatar. So to comprehensively evaluate the effectiveness of the key idea, I strongly suggest extending the model compression to other initialization networks (e.g. [1]) and perform a thorough evaluation under their experimental settings including used datasets, metrics et al.
+ For Table 1, more experimental details might be needed to verify the comparison fairness and validity of the proposed method. For example, what does "MACs" mean? It also looks that the reported scores are different from the scores in Table 3 of PointAvatar [2] such as the rendering time.
+ There are no comparisons for running time. Additionally, the developed method takes 7 hours for training which is somewhat longer than the training time of PointAvatar [2]. Plus the method complexity and other trivial empirical improvements, I would like to say this submission might be lower than the bar of ICLR.
+ No visual comparison results for the ablation studies.
+ In Figure 5, it seems that the results of PointAvatar are more similar to the ground truth than the proposed method. Respecting this point, the performance of the proposed method needs to be significantly improved.
+ In **Abstract** part, please show the full name instead of the abbreviation (e.g. RandLA-Net and GCN) for the first time.
+  There are a lot of typos in the submission. For example, in the **Introduction** part, "In digital avatar reconstruction" --> "in digital avatar reconstruction" && "Inspired by the GAN" --> "inspired by the GAN". Please carefully proofread the main text.

[1] Implicit Neural Head Synthesis via Controllable Local Deformation Fields, CVPR 2023

[2] PointAvatar: Deformable Point-based Head Avatars from Videos, CVPR 2023

**Questions:**

+ In Figure 5, why isn't the background color of "Ours" white, like other baselines? Are all comparisons performed under the same settings?
+ Do you have any ablation study examples to show the importance of "RandLA-Net and GCN layers"?
+ In Table 2-3, the proposed method seems to be significantly better than PointAvatar and the ablation models (although I have some doubts about the correctness of the results). But on the other hand, the qualitative improvements in Figure 4-5 are minor. Can the authors offer some explanations for why your method can produce better scores but with similar visual images?
+ There are other solutions to compress models, so why do the authors choose the knowledge distillation (KD). Can you provide more insights or analysis for it?
+ In the differential renderer part, why do you choose to train GAN. Could other modern generative models, like stable diffusion, help?

---

### Official Review · Reviewer_LZcv · 2023-10-27

**Soundness:** 2 fair
**Presentation:** 3 good
**Contribution:** 3 good
**Rating:** 3
**Confidence:** 4

**Summary:**

This paper introduces a two-stage model compression approach designed to generate high-quality avatar images from point clouds. The first stage enhances feature extraction by integrating point-cloud features with image features, while the second stage employs GAN inversion for producing photorealistic images.  It is reported that this method reduces the number of model parameters while improving rendering quality.

**Strengths:**

1. The motivation is evident, as there is a clear and significant need to develop a lightweight and efficient model for avatar creation.
2. According to reports, high-quality results have been attained using a compact student model.

**Weaknesses:**

1. The paper faces significant issues in its writing, making it hard to follow. These issues encompass the following:
    * Chapter Structure: The paper lacks clarity in explaining how the teacher network is selected and the reason behind each step, as well as the differences between the student and teacher networks. Additionally, the beginning of Sec.3 fails to provide a proper overview of the method and instead delves into implementation details, making it challenging to follow the subsequent content. Furthermore, the final chapter only contains a placeholder.
    * Inconsistent Detail Descriptions: Discrepancies exist between abbreviations and symbols used in Figure 3 and their corresponding captions, while the descriptions in the main text do not align with the figures. Moreover, there are inconsistencies in font usage for formulas within the paper, for instance, the inconsistent appearance of `P` and `Q` in Eq. 3 and 4.
    * Writing Conventions: Several issues with writing conventions are apparent, including 1) Confusingly mixing code and symbols in the descriptions of various network modules, such as S_PointAvatar. 2) Not provide full names or references for many phrases when they are introduced for the first time. 3) Lack of punctuation at the end of formulas and a lack of uniformity in this regard. 4) Figure 2 is not referenced in the main text and lacks a reasonable explanation. 5) Numerous typos, grammatical errors, and descriptive inaccuracies, such as the sentence above Eq. 3, which contains two `:`, and the mention of "super-segmentation" should be "super-resolution" in Sec. 3.2.
2. The novelty of the method is incremental. The first stage combines common techniques from knowledge distillation, while the second stage predominantly relies on StyleGAN-2.
3. There are several confusing elements in the experimental setup. For example, there is no introduction to the structural and parametric differences between the Student and Teacher networks, making the results in Table 1 less persuasive. Additionally, it is unclear why super-resolution is necessary in the first phase and whether it introduces additional errors.

**Questions:**

1. Why were RandLA-Net and EdgeConv, two relatively early methods, chosen as the teacher model? Are there better alternatives available?
2. What are the specific designs for Render G and the inversion network R?
3. How are the computational resources allocated for the second stage of the network?
4. In Figure 5, why do the GT and results from other methods lack background, while the proposed method exhibits different backgrounds?

---

### Official Review · Reviewer_qfwg · 2023-10-28

**Soundness:** 2 fair
**Presentation:** 3 good
**Contribution:** 2 fair
**Rating:** 5
**Confidence:** 4

**Summary:**

This paper aims to efficiently create realistic and animatable head avatars. To this end, the authors propose a two-stage method by first compressing the model via the proposed distillation architecture and then improving the realism by a generative adversarial renderer. Experiments show more realistic results compared to recent methods while reducing the parameters and computation of the network.

**Strengths:**

Overall, this paper presents a feasible solution to create avatars in a more efficient manner. The proposed method is overall technically sound with  state-of-the-art results. From the qualitative comparison shown in Fig. 5, we can that the proposed method has more realistic details.

**Weaknesses:**

There are some concerns regarding the proposed methods.

- The proposed method learns a much larger Teacher Model based on PointAvatar during training, meaning that the computational cost is only reduced for the inference stage of the avatar. It seems that there are some over-claims about the efficiency in the introduction.

- It is not clear why the GCN network is necessary. There is a lack of enough ablation studies on the usage of GCN. It is not easy to draw conclusions from Fig.6.

- The novelty of the StyleGAN-based generative renderer is limited as similar strategies have been used in previous avatar generation methods such as ANR and StyleAvatar.

ANR: Articulated Neural Rendering for Virtual Avatars
StyleAvatar: Real-time Photo-realistic Portrait Avatar from a Single Video

- There is a lack of video results. It is hard to assess the overall quality of the avatar produced by the proposed method.

**Questions:**

- The proposed method needs to extract the color point cloud by PointAvatar. Does it mean that a PointAvatar model needs to be trained beforehand?

- The text in some figures is not very clear for reading. It is recommended to add reference numbers in Tables 1 and 2 for other state-of-the-art methods.

---

### Official Review · Reviewer_S2Ns · 2023-10-30

**Soundness:** 3 good
**Presentation:** 3 good
**Contribution:** 3 good
**Rating:** 3
**Confidence:** 5

**Summary:**

The authors propose a knowledge distillation framework for animatable point-cloud based facial avatars to improve the memory footprint and inference cost. The framework uses RandLA-Net to compute local features and GCN to compute global features as guidance in the teacher model. A GAN based renderer is used as the student model to synthesize realistic avatars from latent codes. The authors demonstrate a 95% reduction in model parameters at a similar rendering quality for on device uses. Comparisons are made with respect to several recent baselines and state of the art performance is demonstrated.

**Strengths:**

1. **Interesting approach** : Using both local and global level features for distillation of point cloud based representation is an interesting approach. Local features capture the detailed information of the point cloud, while global features capture the overall structure of the point cloud. By combining both local and global features, the distilled representation can be more comprehensive and informative.
2. **Ablations**: The need for each component is motivated by necessary ablations.
3. **Comparisons**: The proposed approach is compared against IMAvatar and PointAvatar which are state of the art point cloud based avatar methods and performance improvements are demonstrated over these frameworks.

**Weaknesses:**

1. **Novelty** : The proposed approach’s novelty is limited to using RandLANet and EdgeConv features to distill a pre-existing network architecture from PointAvatar, using pre-existing distillation losses.  Highlighting the novelty would help appreciate the contributions more. In particular, what is the most challenging part and motivation in trying to combine representations from these pre existing networks?
8. **Qualitative results**: More qualitative results required. Can it still generate controllable poses, expressions and viewpoints as in the original PointAvatar?
5. **Clarity**: There are several typographical and syntactical errors (as indicated in nits below) and the manuscript would benefit from a thorough proof reading and formatting pass.
6. **Reproducibility**: The specifics of the student model are not provided anywhere making it hard to reproduce the proposed framework. Providing details of S_PointAvatar either in implementation section or supplementary would be very helpful.
7. **Preliminaries**: A short section of preliminaries, explaining briefly PointAvatar and the GCN+RandLANet networks would make the exposition more self contained making for an easier read.
2. **Notations**: In section 3.1, should equation 1 read
$ \mathcal{E}_{gcn}(x_i^k) = \mathcal{F}(x \bigoplus x_i^k ; \Theta) $ since F is not applied on a set but a single $x_i^k$ ? The nearest neighbor max pooling is accounted for in equation 2. Clarification about the exact nature of $\mathcal{F}$ would be instructive.
3. **Missing Details** Section 3.1 is unclear and lacks sufficient detail. In particular,
> a. Do x_i refer to the points or the features.
> b. Is the distance to the K closest points being concatenated?
4. **Unclear statements** : There are certain phrases uses that are unclear without further elaboration. For instance, What does the phrase “point-level supervised assisted distillation” mean?
7. **3D to 2D rendering**: It is unclear how the 3D renders are used along with G. In particular, Section 3.3 explains how the latent code z is used to modulate the weights of the rendering network, but the how is the Point cloud used in this stage? More details about how the 3D rendering and the GAN objective are combined would be helpful.
8. **Supplementary materials** : Although not necessary, this work would greatly benefit with a supplementary material with additional architectural details and results. Particularly, results with different viewpoints and expressions.

Nits:
> Cho and Hariaran (repeated twice in intro).
> Realistic avatars have been prevalent … (cite).
> Related works - “disposable avatars”?
> Some citations are inside brackets while some are inline, consistent formatting would be helpful.

**Questions:**

1. Is the student model generalizable ? Can we generate entirely new identities using the latent code in GAN renderer?
2. PointAvatar separates the albedo term and the deterministic renderer can then be used for relighting purposes. How is lighting treated in this framework?
3. Should we train a separate student PointAvatar network for every identity? In essence for a video sequence from a new identity does the full training pipeline involve, training the teacher PointAvatar, the Student PointAvatar and $G$ and $R$? If this is the case, is the benefit only at deploying an avatar, but the training cost for every single avatar becomes higher?